# Stepping Beyond Counts in Recovery of Total Hip Arthroplasty: A Prospective Study on Passively Collected Gait Metrics

**DOI:** 10.3390/s23146538

**Published:** 2023-07-20

**Authors:** Camdon Fary, Jason Cholewa, Scott Abshagen, Dave Van Andel, Anna Ren, Mike B. Anderson, Krishna Tripuraneni

**Affiliations:** 1Epworth Foundation, Richmond, VIC 3121, Australia; camdon.fary@unimelb.edu.au; 2Department of Orthopaedics, Western Hospital, Melbourne, VIC 3011, Australia; 3Zimmer Biomet, Warsaw, IN 46580, USA; jason.cholewa@zimmerbiomet.com (J.C.); scott.abshagen@zimmerbiomet.com (S.A.);; 4New Mexico Orthopaedic Associates, Albuquerque, NM 87110, USA

**Keywords:** total hip arthroplasty, gait assessment, E-rehabilitation, arthroplasty patient monitoring

## Abstract

Gait quality parameters have been used to measure recovery from total hip arthroplasty (THA) but are time-intensive and previously could only be performed in a lab. Smartphone sensor data and algorithmic advances presently allow for the passive collection of qualitative gait metrics. The purpose of this prospective study was to observe the recovery of physical function following THA by assessing passively collected pre- and post-operative gait quality metrics. This was a multicenter, prospective cohort study. From six weeks pre-operative through to a minimum 24 weeks post-operative, 612 patients used a digital care management application that collected gait metrics. Average weekly walking speed, step length, timing asymmetry, and double limb support percentage pre- and post-operative values were compared with a paired-sample *t*-test. Recovery was defined as the post-operative week when the respective gait metric was no longer statistically inferior to the pre-operative value. To control for multiple comparison error, significance was set at *p* < 0.002. Walking speeds and step length were lowest, and timing asymmetry and double support percentage were greatest at week two post-post-operative (*p* < 0.001). Walking speed (1.00 ± 0.14 m/s, *p* = 0.04), step length (0.58 ± 0.06 m/s, *p* = 0.02), asymmetry (14.5 ± 19.4%, *p* = 0.046), and double support percentage (31.6 ± 1.5%, *p* = 0.0089) recovered at 9, 8, 7, and 10 weeks post-operative, respectively. Walking speed, step length, asymmetry, and double support all recovered beyond pre-operative values at 13, 17, 10, and 18 weeks, respectively (*p* < 0.002). Functional recovery following THA can be measured via passively collected gait quality metrics using a digital care management platform. The data suggest that metrics of gait quality are most negatively affected two weeks post-operative; recovery to pre-operative levels occurs at approximately 10 weeks following primary THA, and follows a slower trajectory compared to previously reported step count recovery trajectories.

## 1. Introduction

Total hip arthroplasty (THA) is an effective treatment to alleviate pain and improve physical function in patients with severe hip osteoarthritis (OA). Despite expectations to return to normal physical activities [1], many THA patients remain less physically active than age-matched controls through the first year post-operative [2], and some populations do not exceed pre-operative physical activity levels [3,4]. Physical activity questionnaires have been used to measure recovery following THA but are limited by inaccuracies inherent to self-reporting [5,6]. For example, patients with total joint arthroplasties have been documented to overestimate self-reported physical activity levels by as much as 50% [7], and incorrectly conflate reductions in pain and exertion with improved physical function [8].

Measuring step counts during free-living conditions with pedometers and accelerometers can provide an objective measure of physical capacity and functional recovery following total joint arthroplasty [9,10,11]. When using pre-operative values as a baseline, step counts have been reported to recover within 6–7 weeks following THA [6,12,13,14], with less active patients returning to baseline in as little as 3 weeks [14]. However, awareness of monitoring may limit the interpretation of recovery curves due to the presence of the Hawthorne effect (participant reactivity) during gait analysis [15,16,17]. Previous studies have used specific monitoring periods (i.e.: pre-operative, 6th, and 12th week post-operative) which required pedometers to be mailed to participants, attached to clothing upon waking, and then mailed back to clinicians at the end of the monitoring period [18,19]. Such monitoring provides only a snapshot of a patient’s activity, as meteorological (i.e., inclement weather) or personal events (i.e., familial obligations) may lead to atypical variability.

Smartphones and smart activity trackers that allow for continuous passive collection of step counts have been used to objectively measure recovery following THA [11,13,14], and can provide biofeedback that may enhance clinical outcomes post-operation [6,20]. Though many patients return to pre-operative step counts, range of motion and gait deficits have been reported to persist even 12 months post-THA [21,22,23,24,25]. For example, McRory et al. [22] reported lower peak forces, loading rates, and impulses in the operated compared to unoperated limb in pain-free THA patients at 10.5 months (range 2 to 54 months) post-operative. Similarly, Bhargava et al. [21] reported greater double support time, swing time, step time, and ground reaction forces in the operated limb of pain-free THA patients between 6 and 51 months post-operation.

The volume of studies investigating gait parameters in OA patients via wearable sensors has nearly doubled in the past decade. The majority of these studies have employed multiple inertial measurement units (IMU) placed on the trunk, lower limbs, or a combination of each to assess spatiotemporal gait parameters [26]. Wearable single IMU sensors have been reported to provide clinically relevant information in several diseases, including multiple sclerosis [27], post-operative cardiac surgery patients, patients with Parkinson’s disease, and orthopedic patients [26,28]. However, gait analysis with single IMU outside of the lab typically require patients to walk a pre-specified route (i.e., 70 m) [26], which may increase the risk of bias due to dual task interference [29]. Moreover, the data must be transferred back to the researcher/clinician, transformed, and then analyzed, which increases the logistic burden of studying gait longitudinally or in large cohorts [28]. Thus, while the parameters produced during gait analysis have been used to compare THA patients to controls or the operated to the unoperated limb [21,22,23,24,25], few reports exist that compare pre- to post-operative gait parameters over time to establish a recovery trajectory. Additionally, high day to day variations in gait data taken from hip OA patients [30] and differences in gait parameters between measures taken in a lab compared to the field [31] further necessitate the need for frequent data collections during normal ambulatory conditions to develop gait recovery trajectories following THA.

Algorithmic advances and sensor data passively collected by commercial smartphones are now capable of providing metrics of gait quality (i.e., step length, walking speed, double support time, and asymmetry percentage) during activities of daily living [32]. Smartphone-based IMU data has been used to develop predictive models for adverse post-operative events and frailty detection based on gait in cardiac surgery patients [33,34] and assess abnormal gait in osteoarthritis patients [35,36,37]. More recently, passively collected data via a smartphone-based care management platform was used to describe the recovery trajectory of gait metrics following total knee arthroplasty [38,39,40]. To our knowledge, only one study has used passively collected, smartphone-based IMU data to investigate recovery trajectories following THA. Sato et al. [41] measured PROMs, step count, stairs climbed, walking speed, and asymmetry immediately post-operative and at one, three, six, and twelve months post-operative. Pain and mobility PROMs and step counts increased significantly in the first month post-operative, whereas stairs climbed, walking speed and asymmetry all declined in the first month post-operative and did not exceed pre-operative values until three months post-operative. This disconnect between objectively measured physical function and PROMs of function reported by Sato et al. have been reported elsewhere [8,42,43] in the literature and suggests that objective measures of function, when combined with other metrics of recovery, are more robust than step counts or patient reported outcome measures (PROMs) alone.

While Sato et al. [41] demonstrated that gait quality metrics are negatively affected by THA through the first month and exceed baseline by the third month post-operative, we are unaware of any studies that have continuously monitored gait quality recovery during the early post-operative period. Furthering the body of knowledge surrounding the recovery of physical function may benefit surgeons and patients and allow for individual-based, targeted interventions by healthcare providers. The purpose of this study was to assess the recovery of walking speed, stride length, double limb support and asymmetry percentage metrics before and 24 weeks after THA.

## 2. Materials and Methods

This was a level II study with prospectively collected data from an ethics approved (WCG IRB # 20182013) global, multicenter prospective cohort study between January 2020 and March 2022 (clinicaltrials.gov: NCT# 03737149). The global study consisted of three phases including a pilot phase, a randomized controlled trial (RCT), and a longitudinal cohort phase. The methods of the pilot and RCT phases have been previously described and the results published [44,45,46]. Primary outcome measures in the longitudinal phase cohort included assessing adverse events during the episode of care, investigating correlatives of discharge disposition, identifying 90-day range of motion correlates, and correlatives of 90-day satisfaction. Secondary outcomes include range of motion, PROMs (Hip Dysfunction and Osteoarthritis Outcome Score for Joint Replacement (HOOS-JR) and EuroQol 5-dimension 5-level (EQ5D-5L)) collected pre-operatively and at one, three, six, and twelve months post-operative, and Apple health kit data (step counts, flights of stairs climbed, gait quality metrics). Select secondary outcomes have been previously reported for the longitudinal cohort [38,41]. The present study reviewed 30 weeks of passively collected gait quality metrics.

To participate in this study patients needed to be at least 18 years of age, own an iPhone (Apple Inc., Cupertino, CA, USA) capable of pairing with the Apple Watch (Apple Inc.) and supporting updates, scheduled for a primary total hip arthroplasty (THA), and capable of walking with minimal assistance (a single walking stick or single crutch) pre-operatively. Exclusion criteria included patients with substance abuse issues, inflammatory arthropathies, patients participating in other clinical studies, and patients requiring simultaneous or staged bilateral hip arthroplasties less than 90 days apart. All patients gave written consent to participate.

Patients were provided with an Apple Watch and the smartphone-based care management app (mymobility^®^ Care Management Platform, Zimmer Biomet, Warsaw, IN, USA). Patients were provided pre- and post-operative education and exercise content and prescribed an at-home-based therapy program standard to the surgical institution’s standard of care through the app beginning at discharge through to 90 days post-operative. The app recorded patient activity and gait parameters, including walking speed, step length, asymmetry percentage, and double support percentage directly into the platform. Walking speed is an estimation of patient’s velocity while walking on flat ground, step length is an estimation of the distance between foot strikes, double support time is an estimation of the percentage of a gait cycle when both feet are in contact with the ground, and walking asymmetry is an estimation of the percentage of time that asymmetric steps occur within a walking bout [47]. Patients were instructed to carry their phones near hip height in a pocket or waist band and wear their watches whenever possible. No specific guidance was provided through the app regarding frequency, intensity, or duration of walking. Data were automatically collected when the patient took 20 steps in a single direction over a flat surface while the smartphone was located near hip height and coupled with the body (i.e., in a rear or side pocket).

Apple Inc. has reported good concurrent validity and absolute error during a six-minute walk test (6MWT) on a pressure mat while carrying an iPhone in the side pocket for walking speed (ICC = 0.93, σerror = 0.09 m/s), step length (ICC = 0.85, σerror = 0.05 m), double support percentage (ICC = 0.65 σerror = 2.91%) and asymmetry percentage (positive predictive rate = 83.4%, false negative rate = 9.8%) with the iPhone [47].

All data are reported as means with standard deviations unless otherwise indicated. Seven days per week was used to group the weekly data, and data collected during the first six weeks pre-operative were averaged to represent pre-operative values. Recovery was operationally defined as when the respective weekly average gait metric was no longer significantly less than pre-operative. The analysis population selection criteria were all patients eligible who consented, unilateral THA, had key demographic data collected, had follow-up data no shorter than 24 weeks post-operative, and had at least one of the involved Health Kit data metrics (gait speed, gait asymmetry, and step length) collected. Pre- and post-operative gait metrics were compared using paired-samples *t*-tests. Statistical analysis was performed using SAS v9.4 (2013, SAS Institute, Inc. Cary, NC, USA). To control for family-wise error rate due to multiple comparisons, significance was set a *p* < 0.002.

## 3. Results

A Strobe flow diagram of all patients (N = 612, Female: n = 329 (53.8%), Age: 60.7 + 10.5 years, BMI: 29.5 + 6.0 kg/m^2^) is presented in Figure 1. Table 1 shows the frequency of the number of days contributing to the weekly average across the entire data set for each gait quality metric.

### 3.1. Walking Speed

Walking speed was 1.00 ± 0.15 m/s at pre-operative, was lowest at week 2 (0.79 ± 0.17 m/s, *p* < 0.001), recovered at week 9 (1.00 ± 0.14 m/s, *p* = 0.04), and then consistently exceeded pre-operative speeds at week 13 (1.03 ± 0.14 m/s, *p* < 0.001, Figure 2).

### 3.2. Step Length

Step length was 0.59 ± 0.07 m pre-operatively, was lowest at week 2 (0.53 + 0.08 m, *p* < 0.001), recovered at week 8 (0.58 ± 0.06 m, *p* = 0.02), and then consistently exceed pre-operative lengths at week 17 (0.60 ± 0.06 m, *p* = 0.001, Figure 3).

### 3.3. Asymmetry Percentage

Asymmetry percentage at pre-operative was 12.4 ± 12.4%, was greatest at week 2 (42.0 ± 32.6%, *p* < 0.001), recovered at week 7 (14.5 ± 19.4%, *p* = 0.046, and then was consistently less than pre-operative at week 10 (10.5 ± 14.7%, *p* < 0.001, Figure 4).

### 3.4. Double Limb Support Percentage

Double limb support percentage was 31.3 ± 1.4% pre-operatively, was greatest at week 2 (32.8 ± 2.0%, *p* < 0.001), recovered at week 10 (31.6 ± 1.5%, *p* = 0.098), and then was consistently less than pre-operative values at week 18 (31.1 ± 1.6%, *p* < 0.001, Figure 5).

## 4. Discussion

Previously, the parameters produced during gait analysis have been used to quantify abnormal gait in THA patients [21,22]. Smartphone technology has now made it feasible to collect and analyze large amounts of gait quality data. However, to demonstrate the early diagnostic potential of gait quality metrics, the parameters produced must be sensitive to pre-operative impairment, respond to interventions that improve mobility, and provide clinically relevant information [48]. Herein, we provide preliminary evidence of the potential for passively collected gait quality metrics to be used toward early recovery analysis. We found that all measures of gait quality were most negatively affected two weeks post-operative, exhibited a rapid recovery in the first six to eight weeks post-operative, reached pre-operative levels by ten weeks, and then were greater than pre-operative values within the first five to six months post-operative.

The patterns of recovery found in this study are consistent with those previously reported for both objective and patient reported measures [49,50]. The data have shown that following THA a dip in function and an increase in pain has been seen concluding in an inflexion point within 1–2 weeks post-operative followed by gradual improvements. However, they contrast with previous studies that have assessed recovery via patient reported outcome measures (PROMs), such as the HOOS in regard to the speed at which they recover to pre-operative levels. On the one hand, we found the fastest rates of recovery occurred in the first 6–8 weeks, which is similar to the trajectories reported for HOOS measures [50,51,52]. On the other hand, metrics of gait quality remained depressed below pre-operative levels for up to 10 weeks and required up to 17 weeks to exceed pre-operative values. This contrasts with several studies that have reported significant increases in HOOS functional domains (activities of daily living (ADL) and sports and recreation) in as little as 8 to 13 days post-operative [53,54]. This supports the need for more precise measures as it is likely that the HOOS domains could be limited by a ceiling effect.

Further, identifying patients vulnerable to inadequate functional recovery can enable timely and effective interventions [55]. As noted, PROMs are limited by ceiling effects [56,57] and language that results in patients over-estimating physical function [43] and incorrectly conflating reductions in pain and exertion with improved physical function [8,58,59]. A mounting body of literature is demonstrating a disconnect between PROMs and physical function, whereby PROMs increase but functional performance tests such as the timed-up-and-go, stair climb, and 6 min walk test remain depressed in the early to mid-post-operative period [53,54,60,61,62,63,64,65,66,67,68]. With regard to gait, Bolkesteijn et al. [51] reported walking speed, stride length, and step time asymmetry returned to pre-operative levels at two months post-operative. However, HOOS functional domains significantly increased in the first two months post-operative, with improvements exceeding the minimally clinically important differences (MCIDs). Similarly, Sato et al. [41] reported the greatest increases in the HOOS subdomain of mobility in the first month post-operative while select functional metrics (stairs climbed, gait speed and asymmetry) decreased in the first month following TKA. Given that physical performance remains lower than pre-operative during the early post-operative period, the findings of Bolkensteijn et al. and Sato et al. in conjunction with ours suggest that gait quality metrics may be a more sensitive determinant of functional recovery.

Although step counts provide an objective measure of physical activity capacity, they do not provide information related to compensations that may affect function or contralateral joint health in the future, nor the intensity or quality of physical activity. Additionally, month to month meteorological variabilities can lead to seasonal differences in step counts of more than 1000 per day [69] and further complicate the assessment of individualized recovery. Our results suggest that gait quality is initially negatively affected by THA surgery, and follows a distinct, but slower recovery trajectory than step counts. For example, in a similar cohort, Sato et al. [41] reported significantly increased steps per day in the first month post-operative while metrics of gait quality remained less than pre-operative. Compared to step counts, continuous monitoring of gait parameters during the recovery from THA may also provide deeper insights into a patient’s ability to perform ADLs or physical recreation. For example, an absolute minimum of 0.49 m/s is necessary to cross a two-lane street with full time allotment and the mean walking speed used by pedestrians at crosswalks is 1.32 m/s [70]. Gait velocity of THA patients has been reported to range from 0.97 to 1.10 m/s at 6 weeks post-operative, and 1.18 to 1.20 m/s at 16 weeks post-operative [71,72,73]. Step length has been reported to range between 0.60 and 0.69 m at 6 weeks and up to 0.77 m 6 months post-operative [71,73]. While these values are slightly higher than those reported in the present study, they are all markedly lower than the walking speeds of 1.3 to 1.41 m/s and step lengths of 0.68 to 0.85 m reported for healthy controls [74]. Strength of the hip abductors, flexors, and extensors, step width, foot strike and toe off angle, and lumbar-pelvic-hip range of motion have been associated with spatial-temporal gait disturbances in THA patients [75,76,77]. As these gait parameters seem to be inherently linked with other gait disturbances that occur in the hip and knee of OA patients, gait speed and step length may be a simple to monitor, but a highly sensitive marker of recovery following THA [51]. Additionally, differences in gait parameters have been reported between walking in a rested condition versus walking following a battery of physical performance tests designed to recreate ADLs in THA patients [78]. Thus, continuous measurement under free-living conditions may increase the ecological validity of using gait quality to monitor recovery following THA.

### Limitations

Evaluating passively collected data from digital care management systems is inherent to certain limitations. This includes patient-specific behaviors associated with carrying their smartphone; for example, if the phone is on an armband, gait quality metrics will not be collected. However, given this is unique to each patient, it seems reasonable that these behaviors would be similar both pre- and post-operatively. The improvements seen that exceeded pre-operative values suggest overall data collection in this population was not affected by patient behaviors. There may be some effects of the study on patient behavior, but the collection of free-living data from the patients’ smartphones likely eliminates the Hawthorne effect resulting in gait evaluations similar to everyday life as opposed to in the lab as seen in traditional gait analyses [15,16,17]. This also supports data collected in real-life settings. However, this study is also limited as the sensor used has not been validated in THA patients. While the sensor has been validated in a population with demographics (age, BMI, prevalence of musculoskeletal disorders) similar to this population [47], it is unclear how gait disturbances following THA may influence its reliability and validity. For example, in the first week post-operative, patients with walkers and assistive devices likely do not have the same recognizable gait patterns as those without assistive devices, leading to a lack of data registration by the sensor device. Additionally, changes in attire immediately after THA, such as gowns, pajamas, and other loose-fitting clothing limit data capture, as the phone may not be properly coupled with the body as a result. These factors that limit data capture by the sensor device likely explain the slightly reduced sample size in the final gait analysis in the present study (Table 1). Further analysis is needed to determine the effect THA, comorbidities, and post-operative adverse events have on gait patterns.

Finally, because we did not report PROMs, we are unable to perform any inferential associations or comparisons between the recovery of gait quality metrics and patients’ perceptions of their functional recovery within this cohort. While PROMs have become an important assessment of performance and value by Medicare and many private health insurers [79], many studies show they lack sensitivity to measure changes in physical function when compared to objective measures [53,54,60,61,62,63,64,65,66,67,68]. Several studies show a stronger correlation between pain and PROM function than PROM function and objective function [8,58,80], likely due to the parallel item content of pain and function on regularly used hip specific PROMs, such as the Oxford Hip Score (OHS) and HOOS [59]. Hip specific PROMs also demonstrate large ceiling effects, with ceiling effects found in 20.8% of patients on the OHS [81], 16% in HOOS, and 36% in HOOS-JR [82]. Moreover, when evaluating HOOS subdomains related to function (sport and ADL), ceiling effects are preset in 28% to 33% of patients [82], and ceiling effects of 39% have been reported for the HOOS 12 item subdomain of function [56]. These aforementioned findings demonstrate the need for more objective measures when assessing the recovery of physical function following THA.

## 5. Conclusions

Monitoring mobility in the home setting has historically suffered from several challenges including accuracy, technology portability, and data accessibility. All four sensor-derived gait quality metrics measured in this study displayed a similar recovery curve and were responsive to THA. The results suggest that gait quality metrics measured in the home setting follows a slower trajectory of recovery compared to PROMs or gait parameters measured in a laboratory. The results of this study demonstrate the feasibility of collecting gait quality metrics via a digital care management platform following THA. Further, the data suggest that recovery to pre-operative levels occurs at ten weeks following primary THA, with post-operative week 2 having the most inferior point along the recovery curve. These trajectories follow a slower pace compared to previously reported step count recovery trajectories. Thus, it may be time to combine gait quality metrics with PROMs and step counts when measuring recovery following joint arthroplasty.

## Figures and Tables

**Figure 1 sensors-23-06538-f001:**
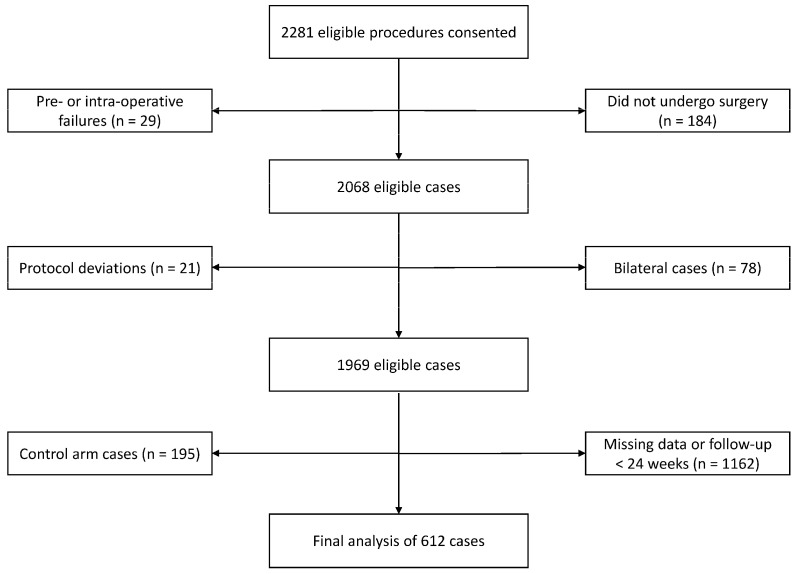
Strobe Flow Diagram.

**Figure 2 sensors-23-06538-f002:**
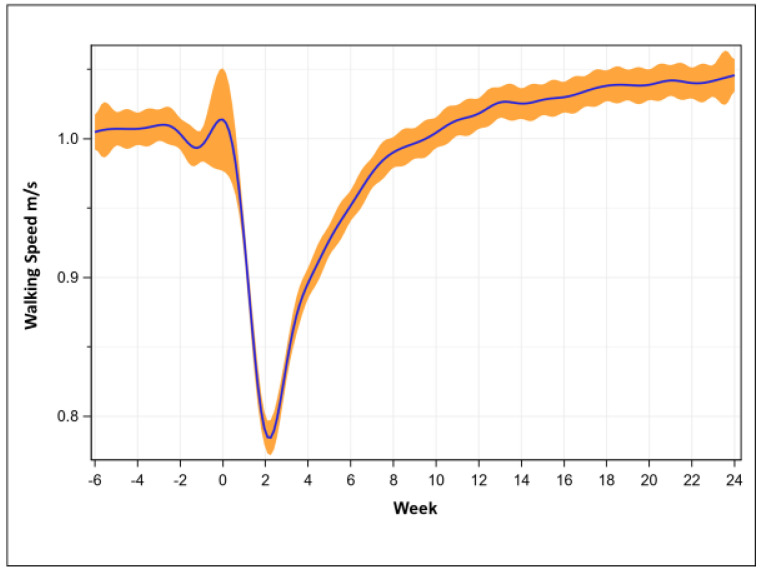
Walking speed recovery trend (mean with 95% confidence intervals, n = 605).

**Figure 3 sensors-23-06538-f003:**
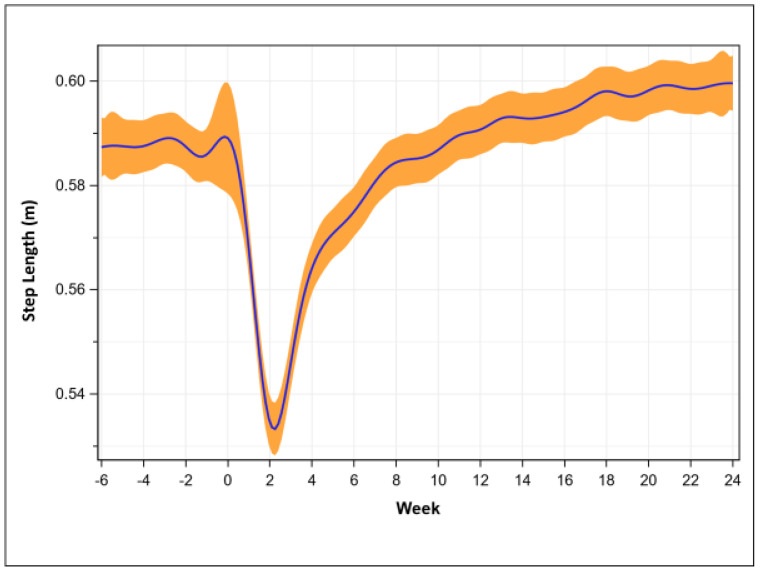
Step length recovery trend (mean with 95% confidence intervals, n = 605).

**Figure 4 sensors-23-06538-f004:**
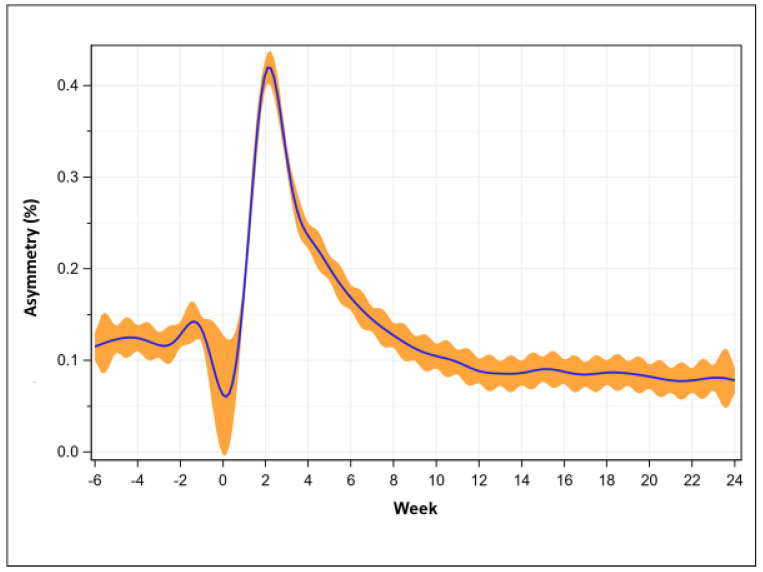
Asymmetry percentage recovery trend (mean with 95% confidence intervals, n = 582).

**Figure 5 sensors-23-06538-f005:**
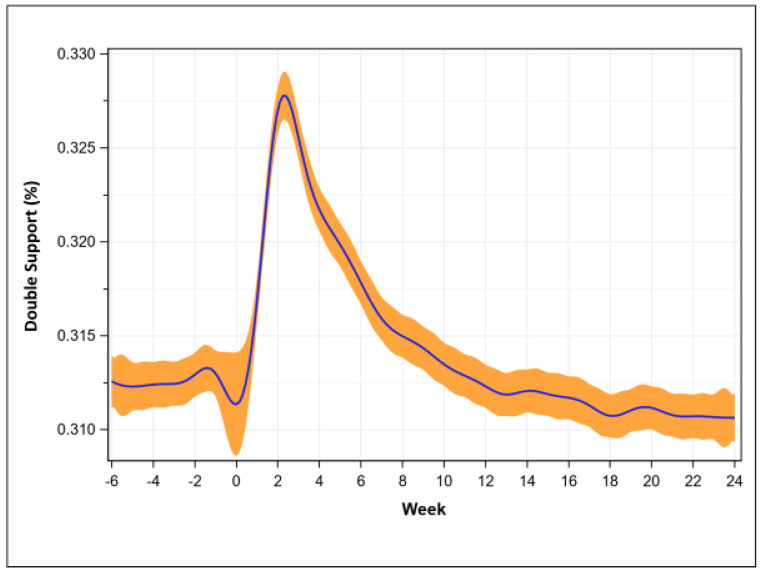
Double limb support percentage recovery trend (mean with 95% confidence intervals, n = 597).

**Table 1 sensors-23-06538-t001:** Frequency of days per week contributing to the weekly average in the data set.

Number of Days	Gait Speed	Asymmetry	Double Support	Step Length
	%	%	%	%
1	2.4%	7.5%	3.6%	2.4%
2	3.2%	8.1%	3.9%	3.2%
3	3.8%	9.8%	4.5%	3.8%
4	5.6%	12.2%	6.4%	5.6%
5	8.5%	15.5%	9.3%	8.5%
6	17%	19.7%	17.8%	17%
7	59.4%	27.3%	54.5%	59.3%

## Data Availability

The data are unavailable.

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
