# Peer review of "Stepping Beyond Counts in Recovery of Total Hip Arthroplasty: A Prospective Study on Passively Collected Gait Metrics"

_sensors, 2023, doi:10.3390/s23146538_

Round 1

Reviewer 1 Report

This paper provides a longitudinal examination of pre and post-operative gait metrics in THA using commercial wearables. The paper is well written, and would be of interest to researchers and medical professionals working in this area. However, the work and paper have some significant limitations as itemized below.

Introduction

-The stated goal of the work “The purpose of this feasibility study was to evaluate pre- and post-operative gait quality data to assess recovery following primary THA” is really much too broad and needs to be stated with purpose (i.e. what is the purpose of evaluating pre and post gait data?)

-Assuming that one of the goals of this paper is to demonstrate the use of a wearable gait assessment protocol to assess recovery longitudinally, then the background/introduction section needs to provide the relevant background on this topic. Currently, there is no mention of the research that is being done in wearables and gait measurements (THA or other medical areas etc)

Methods

-line 110 – error for walking speeds seems to be very large at 0.9m/s

-Line 105 to 12 – is this performance data from THA patients? If it is not then there is a significant concern here. Validation of the performance of the wearable systems would need to be done first, on THA. For example, one might want to do a direct comparison of the wearable gat data to the gait lab data in THA. The data in this study does not provide evidence that the wearable gait metrics are accurate or reliable when used in THA.

-lines 106 and 206+ – why is the patient activity data (step counts) not analyzed and reported here?

-provide equation for asymmetry

- line 122 - why only one gait metric? If that was the case that there were missing data, this needs to be clearly presented in the results. Provide summary information about what gait parameters were collected for each participant, and in the graphs provide the n’s.

-the use of the paired t-test seems rather basic (and perhaps insufficient) for the determination of inferiority. Consider consulting a statistician.

-Also, were the data normally distributed?

Results

-The figures need to illustrate the standard deviations and note in the captions that the data represents the means. The statistically significant events can also be shown on the graphs.

-Figure 4 – what is the reason for the dip at week = 0?

Discussion

-The authors discuss the results of the gait metrics with respect to published literature that used other (self-report, functional etc) outcomes to assess THA recovery. This clinical trial collected a number of other relevant outcome measures such as TUG, HOOS, EQ-5D etc) and it is not clear why these are not mentioned in the methods and  being reported here to show how the gait metrics compare and correlate. As it stands the paper is rather weak, as it does not directly compare to any other established and gold-standard outcome measures for THA, or active gait analysis as noted previously.

-line 208 – Why not compare to step counts measured in this study?

-line 213 - provide a range for 1.19 m/s

-Explain the Hawthorne effect

Conclusions

-        “All four sensor derived gait quality metrics measured in this study displayed a similar recovery curve and were responsive to THA.” Given the limitations stated above, this is a lofty claim. The paper can be much improved by expanding the analysis and comparing the gait results to other recovery related metrics.

-        “The results of this study demonstrate the feasibility of collecting gait quality metrics via a digital care management platform following THA” – Need to provide a discussion of other literature related to work using wearables for gait assessments within or even outside of THA.

Quality of writing is adequate

Author Response

Reviewer 1:

This paper provides a longitudinal examination of pre and post-operative gait metrics in THA using commercial wearables. The paper is well written, and would be of interest to researchers and medical professionals working in this area. However, the work and paper have some significant limitations as itemized below.

Thank you for the positive feedback.  We have worked to bolster the background information to better describe the need for this study.  We have also updated our purpose statement and title according to the reviewer’s critique. The main purpose of this study was to demonstrate the ability to use sensor-based devices as objective measures that provide details of the recovery of physical function that activity levels alone do not.

To strengthen our paper, we have increased the sample size from n=395 to n=612.  We were able to do this because while the initial inclusion criteria of the study were limited to Zimmer Biomet devices, a subsequent amendment the application was made device agnostic. We have updated the patient attrition chart accordingly.  Please note that while the sample size has increased, the results have not changed.

We hope you find the following responses and corrections to our manuscript satisfactory. 

-The stated goal of the work “The purpose of this feasibility study was to evaluate pre- and post-operative gait quality data to assess recovery following primary THA” is really much too broad and needs to be stated with purpose (i.e. what is the purpose of evaluating pre and post gait data?)

We have changed the purpose statement to be more specific to the objectives of this study.  The purpose now states:  “The purpose of this feasibility study was to assess the recovery of walking speed, stride length, double limb support and asymmetry percentage metrics before and 24 weeks after THA.”

“The purpose of this prospective study was to observe the of physical function following THA by assessing passively collected pre- and post-operative gait quality metrics.

-Assuming that one of the goals of this paper is to demonstrate the use of a wearable gait assessment protocol to assess recovery longitudinally, then the background/introduction section needs to provide the relevant background on this topic. Currently, there is no mention of the research that is being done in wearables and gait measurements (THA or other medical areas etc)

We agree and have added information found in lines 70 through 107.  Thank you for this feedback, we believe it has strengthened the case to conduct this study.

 -line 110 – error for walking speeds seems to be very large at 0.9m/s

Thank you for this astute observation. This was a typographical mistake, the error reported by apple was 0.09 m/s and has been updated.

-Line 105 to 12 – is this performance data from THA patients? If it is not then there is a significant concern here. Validation of the performance of the wearable systems would need to be done first, on THA. For example, one might want to do a direct comparison of the wearable gat data to the gait lab data in THA. The data in this study does not provide evidence that the wearable gait metrics are accurate or reliable when used in THA.

This validity and reliability data was not exclusively from THA patients.  However, the demographics of the subjects in the performance study are representative to those in our study. Those subjects were older (~75 years of age, range 64 to 92)) and 81% of the subjects suffered from a musculoskeletal disorder.  Of the subjects that reported a musculoskeletal disorder, 49% had osteoarthritis, 26% reported “other arthritis” and 16% reported a joint replacement surgery.  The subject pool was not comprised of young adults without osteoarthritis or other musculoskeletal complications. Because the sensors were accurate and reliable in patients with musculoskeletal disorders, and because the benchmark for recovery in the present study was pre-operative (patients with musculoskeletal disorders, specifically OA), we see no mechanism whereby THA surgery would affect the accuracy and reliability of the gait metrics.

-lines 106 and 206+ – why is the patient activity data (step counts) not analyzed and reported here?

The primary purpose of this study was to describe the recovery trajectory of gait quality metrics following THA. As mentioned in the introduction (lines 53-55 of the revised manuscript), step counts display high variability and are affected by meteorological variance. Additionally, step counts in older adults show considerable seasonal variation (https://pubmed.ncbi.nlm.nih.gov/28831368/: greater step counts in spring and fall versus winter and summer), which further complicates any comparison.

-provide equation for asymmetry

Detailed information about the biomechanical model and algorithms used to estimate asymmetry and the other gait parameters have not yet been published by Apple Inc.

- line 122 - why only one gait metric? If that was the case that there were missing data, this needs to be clearly presented in the results. Provide summary information about what gait parameters were collected for each participant, and in the graphs provide the n’s.

One day per week and one gait metric was the minimum requirement, as many patients only had data for one day or one gait metric in the first week post-operative.  The frequency of number of days contributing to the weekly average across the entire data set for each metric is as follows:

Gait speed

Asymmetry

Double Support

Step Length

Number of Days

%

%

%

%

1

2.4%

7.5%

3.6%

2.4%

2

3.2%

8.1%

3.9%

3.2%

3

3.8%

9.8%

4.5%

3.8%

4

5.6%

12.2%

6.4%

5.6%

5

8.5%

15.5%

9.3%

8.5%

6

17%

19.7%

17.8%

17%

7

59.4%

27.3%

54.5%

59.3%

-the use of the paired t-test seems rather basic (and perhaps insufficient) for the determination of inferiority. Consider consulting a statistician.

We have consulted with our statistician. The paired t-test is a well-accepted statistical testing method for comparing the mean difference of two samples in the form of matched pairs in the univariate test context with a defined or involved inferiority margin. If the reviewer meant to do true inferiority or superiority or non-inferiority test by “the determination of inferiority” with specific margins, the same test method could be applied but with different null values; however, as this was an preliminary analysis without an inferiority margin previously defined in the literature, we believe the paired t-test is the most appropriate analyzation method.

-Also, were the data normally distributed?

Even though as a parametric assumption for the paired t-test, the paired measurement difference between the two samples should be normally distributed, the paired t-test is robust to the non-normal data and is valid when there are more than 20 – 25 data records/observations per group.  This is a well-accepted statistical practice. Paired t-test and t-test in general works well even for skewed distribution when sample is large enough. The sample sizes in this analysis are large enough to ease the normality concern, and therefore no normality test was done in this analysis.

Results

-The figures need to illustrate the standard deviations and note in the captions that the data represents the means. The statistically significant events can also be shown on the graphs.

We agree and have added the 95% confidence interval band to each figure.

-Figure 4 – what is the reason for the dip at week = 0?

We are unsure which dip the reviewer is referring to (the one prior to the operation (week 0) or the one that follows), and so will address both. 

Prior to week 0: Please note that all figures show a positive inflection point (increase in asymmetry and double support percentage and decrease in speed and length) the week prior the operation, followed by a return to closer to the pre-operative average the week of the operation. We discovered this same pattern in TKA patients in our previous study: https://www.mdpi.com/1424-8220/23/12/5588. We speculate this difference was likely due to a change in behaviors as patients sought to get their affairs in order prior to the surgery but have no way of verifying this speculation.

Following week 0: This pattern of an immediate drop is consistent with prior publications on recovery following THA for both objective (Luna et al. 2019: https://pubmed.ncbi.nlm.nih.gov/30039461/) and patient reported outcome measures (Goeb et al, 2021: https://pubmed.ncbi.nlm.nih.gov/33840540/). Given the surgical trauma an immediate drop in function and increase in pain is expected. After the initial dip, this gradually improves as expected. This has been clarified in the discussion’s second paragraph beginning on line 213.

-The authors discuss the results of the gait metrics with respect to published literature that used other (self-report, functional etc) outcomes to assess THA recovery. This clinical trial collected a number of other relevant outcome measures such as TUG, HOOS, EQ-5D etc) and it is not clear why these are not mentioned in the methods and being reported here to show how the gait metrics compare and correlate. As it stands the paper is rather weak, as it does not directly compare to any other established and gold-standard outcome measures for THA, or active gait analysis as noted previously.

There were three cohorts in the mymobility study, and data in this study came from Phase II and Phase III. Timed up-and-go and the single-leg-stance were only collected in the randomized clinical trial (phase II), thus we cannot include those analyses in this report, as they would represent only a small percentage of total subjects.

In regard to the PROMs, these were collected pre-operatively and at one, three, six, and twelve months post-operatively.  These results have been reported elsewhere (Sato et al. 2023). Sato et al. was published after our paper was submitted and we have incorporated a discussion of the findings into the introduction (Line 103-112) and discussion (Lines 240-246) of the present paper. Thus, while we can compare our gait quality metric findings to the reported PROM findings, we cannot report them in this study nor perform any quantitative analysis of them.

In response to the weakness of the study, we would like to refer the reviewer to our paper on TKA that was recently published in Sensors incorporating the same outcome variables as the present study: https://www.mdpi.com/1424-8220/23/12/5588

-line 208 – Why not compare to step counts measured in this study?

Please see the previous answer to this query from the methods.

-line 213 - provide a range for 1.19 m/s

Done. The range reported was 1.18 to 1.20 which is why we listed 1.19 initially.

-Explain the Hawthorne effect

 We have provided citations that demonstrate the presence of a Hawthorne effect when measuring gait in a laboratory and included the more recently used term “participant reactivity” in parenthesis.

-       “All four sensor derived gait quality metrics measured in this study displayed a similar recovery curve and were responsive to THA.” Given the limitations stated above, this is a lofty claim. The paper can be much improved by expanding the analysis and comparing the gait results to other recovery related metrics.

We respectfully disagree.  All the gait metrics showed a similar pattern: they were all significantly negatively affected by the surgery in the first two weeks post-operative, they all started to improve following the 2nd week post-operative, and then they all recovered to pre-operative values within 8-10 weeks post-operative. While this recovery trajectory is different from those previously reported for PROMs or step counts, this statement does attempt to draw a comparison, but rather simply presents the recovery of these four metrics.

However, given the reviewers concerns, we have added the lack of PROMs into the limitations section, found at lines 289-305.

-        “The results of this study demonstrate the feasibility of collecting gait quality metrics via a digital care management platform following THA” – Need to provide a discussion of other literature related to work using wearables for gait assessments within or even outside of THA.

We have now provided previous sensor work related to THA and outside THA in the introduction.

Reviewer 2 Report

INTRODUCTION

·       Consider replacing “total joint arthroplasty patients” by “patients with total joint arthroplasty”

·       I recommend developing this argument, as it is not clear or well justified: “The lack of longitudinal data is likely because collecting gait characteristics was historically performed in a lab and required a force plate and/or multi-camera motion analysis systems. Additionally, high day to day variations in gait data taken from OA patients [23] and differences in gait parameters between measures taken in a lab compared to the field [24] further limit the development of gait recovery trajectories following THA.”

·       Please, give examples of “gait quality” metrics. What do you refer to here?

·       Give examples here and justify this argument: “when combined with other metrics of recovery, are more robust 79 than step counts, or patient reported outcome measures (PROMs) alone”

·       The methodology includes a therapy provided online. Why is not mentioned in the introduction and not justified?

METHODS

·       How was the sensor located near the center of mass? Was this instructed to the patients?

RESULTS

·       Consider including a table with a summary of the clinical and demographic information of the patients.

DISCUSSION

·       Consider referring to the confounding effect of contextual factors, as indicated in the introduction.

·       Consider indicating that the position of the Apple Watch is not the most favourable for biomechanical characterisation and the technical validity of the measures may affect the results.

·       Consider indicating that although patient reported outcome measures (PROMs) lack of accuracy, reliability, etc. these are considered the gold standards. Thus, a limitation of this study is that no associations have been studied between the PROMs and the gait measures reported here.

·       The first paragraph (the first half) is repeating the arguments from the introduction, consider summarising it, as the second half of this paragraph is the most relevant.

·       The effect of the therapy is not mentioned in the discussion.

The comments are indicated in the previous section

Author Response

Reviewer 2:

We appreciate your time and thoughtful review to improve our manuscript.  We hope you find the responses and manuscript corrections satisfactory.

  Consider replacing “total joint arthroplasty patients” by “patients with total joint arthroplasty”

This was changed at line 40-41

  • I recommend developing this argument, as it is not clear or well justified: “The lack of longitudinal data is likely because collecting gait characteristics was historically performed in a lab and required a force plate and/or multi-camera motion analysis systems. Additionally, high day to day variations in gait data taken from OA patients [23] and differences in gait parameters between measures taken in a lab compared to the field [24] further limit the development of gait recovery trajectories following THA.”

We agree this will strengthen the paper.  In conjunction with the comments from reviewer 1, we have added additional information that we believe strengths and further develops this statement. Please see lines 70-87 and 109-114.

  • Please, give examples of “gait quality” metrics. What do you refer to here?

Please see lines 80-91.  Further information has been added in the methods at lines 140-144.

  • Give examples here and justify this argument: “when combined with other metrics of recovery, are more robust 79 than step counts, or patient reported outcome measures (PROMs) alone”

Please see lines 96-107.

  • The methodology includes a therapy provided online. Why is not mentioned in the introduction and not justified?

The present study is a secondary analysis of prospective data with the purpose to describe the recovery from THA with gait quality metrics. The effects of traditional care versus care delivered via a smartphone-based care management platform are outside the scope of this study and have already been reported in these cohorts (see references 44-46). Given the purpose of the study, and since all patients in the present study received the same digital-based therapy, we do not believe there is anything that needs to be discussed besides the description in the methods.

  • How was the sensor located near the center of mass? Was this instructed to the patients?

Please see lines 137-149.

  • Consider including a table with a summary of the clinical and demographic information of the patients.

We believe the current form is most appropriate. There are only 3 demographic variables, which does not lend itself very well to a table.  On the other hand, if by clinical the reviewer means the gait quality metrics, there are 25 rows for each of the four variables, which we think would clutter the paper and is better represented in the figures. To make the figures more descriptive, we have added the 95% confidence intervals.

  • Consider referring to the confounding effect of contextual factors, as indicated in the introduction.

We have added further information about the contextual factors into the limitations section.

  • Consider indicating that the position of the Apple Watch is not the most favourable for biomechanical characterisation and the technical validity of the measures may affect the results.

The gait metrics reported in this study were not collected by the Watch but instead by the inertial measurement units found within the phone.  We hope lines 144-149 clarify this.

  • Consider indicating that although patient reported outcome measures (PROMs) lack of accuracy, reliability, etc. these are considered the gold standards. Thus, a limitation of this study is that no associations have been studied between the PROMs and the gait measures reported here.

We have added this information into lines 290-307

  • The first paragraph (the first half) is repeating the arguments from the introduction, consider summarising it, as the second half of this paragraph is the most relevant.

We have removed some text from the first half of the paragraph.

  • The effect of the therapy is not mentioned in the discussion.

Evaluating the therapy provided by the app was not the purpose of this study.  Patients received therapy exercises via the app in lieu of in person physical therapy, however, the exercises were standardized to each institution/study site’s standard of care and not specific to each patient. There was also no control group that received traditional therapy to compare any effects to. Therefore, there was no effect of therapy to measure, nor do we see how standard of care therapy is relevant to the findings or necessitates discussion.

Round 2

Reviewer 1 Report

The authors have made some substantial improvements to the paper. However, there are still a number of important issues to be addressed, and some new ones from their responses.

The authors have now clarified the reason for why the one day per week and one gait metric was the minimum requirement and provided a table to show what data were collected, as many patients only had data for one day or one gait metric in the first week post-operative. What is the reason for this? One would expect a lot more data given that the only requirement was 20 steps on level ground.  Please provide a clear explanation for why continuous monitoring resulted in so few data points for so many participants. This is important information that might be useful for others intending to use this type of remote monitoring. Also, the table that was provided in the response showing what data was collected needs to be included in the paper itself.

The concern with using the Apple wearable on a patient population that it has not been validated on, needs to be better acknowledged as a limitation. These wearables use certain algorithms that may or may not work well on individuals with certain gait deviations. The authors admit that they do not know what algorithms Apple uses, so this is a concern/limitation.

The authors claim that step counts are not a good measure, but provide only one reference to support this. Why not analyze the step count data here to provide more solid evidence here?

Regarding the statistics, a post hoc adjustment such as the Bonferroni should be applied because of the increased risk of a type I error when making multiple statistical tests (i.e. multiple gait parameters). A more suitable p value would be around 0.01. Please revise.

Regarding my earlier comment about including the other outcome measures in the analysis, this manuscript needs to more clearly explain the entire study, including the study phases, outcome measures, and other publications that are based on it. It must be clear what the entire study involved, and what portion of data are being presented here.

please see comments to authors

Author Response

The authors have made some substantial improvements to the paper. However, there are still a number of important issues to be addressed, and some new ones from their responses.

The authors have now clarified the reason for why the one day per week and one gait metric was the minimum requirement and provided a table to show what data were collected, as many patients only had data for one day or one gait metric in the first week post-operative. What is the reason for this? One would expect a lot more data given that the only requirement was 20 steps on level ground.  Please provide a clear explanation for why continuous monitoring resulted in so few data points for so many participants. This is important information that might be useful for others intending to use this type of remote monitoring. Also, the table that was provided in the response showing what data was collected needs to be included in the paper itself.

We have added the table to the results section of the paper.

The reduced sample size was a human error. Removing the exclusion of non-Zimmer Biomet devices increased our sample size from 395 to 612.  However, the output tables did not populate with the additional data, and this is why the sample size of each gait metrics is in the upper 300s. We have reran the code and produced accurate tables, and the sample size for each gait metric is now accurate (between 582 and 605).

The increase in sample size had a small effect on recovery (shifting forward in some cases by one week) on when values surpassed pre-operative (shifting forward by 1 to 5 weeks).

Even so, the reviewer brings up a good point regarding the data capture limitations with smartphone behavior that likely explain the reduced sample size upon final gait analysis.  We have included this information in the limitations in lines 313-321.

The concern with using the Apple wearable on a patient population that it has not been validated on, needs to be better acknowledged as a limitation. These wearables use certain algorithms that may or may not work well on individuals with certain gait deviations. The authors admit that they do not know what algorithms Apple uses, so this is a concern/limitation.

Please see lines 310-313 in the limitations.

The authors claim that step counts are not a good measure, but provide only one reference to support this. Why not analyze the step count data here to provide more solid evidence here?

At no point in the manuscript do we claim step counts are not a good measure, nor was the purpose of this paper to discount the utility of step counts. However, As there are no universally accepted objective measures for functional recovery in day to day living, we would be remiss not to evaluate all aspects of gait metrics available using this sensor-based instrument and to report on the recovery trajectories available. The main message we have tried to convey is step counts are not a complete measure of recovery. To this point, we state at multiple points in the manuscript that step counts provide an objective measure of recovery before listing step count limitations: Lines 45-47 and 268-270.  Step count data from the longitudinal cohort was previously published, albeit at one-, three-, six- and twelve-months post-operative, and have cited and drawn comparisons to this report in the manuscript (line 274-277). We can see how the last line of the conclusion may have given this impression, and have changed our final concluding line to:

“Thus, it may be time to combine gait quality metrics with PROMs and step counts when measuring recovery following joint arthroplasty.”

Regarding the statistics, a post hoc adjustment such as the Bonferroni should be applied because of the increased risk of a type I error when making multiple statistical tests (i.e. multiple gait parameters). A more suitable p value would be around 0.01. Please revise.

This is a valid concern. We made two main comparisons per gait metric: When did the post-operative value no longer differ from pre-operative (recovery) and when was the post-operative value significantly superior to pre-operative (greater for speed and length, less for asymmetry and double support).  The most straight forward method to account for this is to multiply the p-values together, which gives up a p value of < .0025, which agrees with the Bonferroni correction of p < .002083 and the Sidak correction of p < .002135.  To be conservative and for simplicity, we have set the significance as anything less than p = .002.  This has had small changes on our findings: recovery was shifted up by one week and superiority was shifted later in the post-operative period by 1-3 weeks.  

Please see lines: 20-26, 29, 176-177, 196-198, 202-204, 208-210, 215-217, 213, and 242-243

Regarding my earlier comment about including the other outcome measures in the analysis, this manuscript needs to more clearly explain the entire study, including the study phases, outcome measures, and other publications that are based on it. It must be clear what the entire study involved, and what portion of data are being presented here.

We have distinguished the three phases of the study registered on Clinicaltrials.gov, explained the primary and secondary outcomes of the third phase (where the data from this study was derived from) and appropriately cited publications that have used data from all three phases.  Please see lines 120 through 131.
